# A Study on the Soundscape Preferences of the Elderly in the Urban Forest Parks of Underdeveloped Cities in China

**Lei Luo** [1,†], **Qi Zhang** [1,†], **Yingming Mao** [2], **Yanyan Peng** [3], **Tao Wang** [4] **and Jian Xu** [1,5,6,*]

1 Department of Tourism Management, South China University of Technology, Guangzhou 511442, China; luol19991124@163.com (L.L.); zhangqiqi3590@126.com (Q.Z.)
2 Graduate School of Horticulture, Faculty of Horticulture, Chiba University, Chiba 271-8510, Japan; skyisyui@gmail.com
3 Department of Urban Planning and Management, School of Public Administration and Policy, Renmin University of China, Beijing 100872, China; pengyanyan2018@ruc.edu.cn
4 Zhijiang College, Zhejiang University of Technology, Shaoxing 312000, China; wt@zzjc.edu.cn
5 Key Laboratory of Digital Village and Sustainable Development of Culture and Tourism, Guangzhou 510006, China
6 Tourism Strategy and Policy Research Center, Guangzhou 510006, China
* Correspondence: jianxu@scut.edu.cn; Tel.: +86-181-4288-6885
† These authors contributed equally to this work.

**Abstract:** Against the backdrop of the global aging trend, the proportion of the elderly population is severely increasing in the urban areas of underdeveloped regions. Despite evidence that urban forest parks are effective at enhancing the physical and mental well-being of the elderly, little has been done to investigate the connection between urban forest parks and the elderly in underdeveloped regions, and landscape studies in particular are lacking. This study attempted to address this gap, using a subjective evaluation method in which 725 elderly respondents were engaged in a questionnaire survey on their soundscape preferences in the urban forest parks of an underdeveloped city in China. The results revealed the elderly people's preferences for soundscapes, and a further analysis demonstrated the relationships between these preferences and landscape features. The effects of personal traits and living situations on soundscape preferences were determined by analyzing the impacts of living conditions, occupation, and education on soundscape preferences. By building a model with regression coefficients, the most powerful factors influencing soundscape choice were investigated. It was found that (1) the types of sound sources preferred by the elderly, in descending order, were natural sound, livestock sound, bird song, musical sound, other sounds. (2) The differences among education, occupation, and age all affected the participants' soundscape preferences, i.e., the mean values of the soundscape preferences among older adults varied with education, occupation, and age. The mean value of soundscape preference was higher among older adults who had received higher education, were government officials and business managers, and belonged to higher age groups. (3) Among the various factors influencing the soundscape preference of the elderly, the most influential factors were the length of time spent in the waterfront environment, the time spent in the forest park, and the importance of road signs. (4) The preference for soundscapes was strongly connected with happiness in life. (5) Wearing a mask significantly reduced soundscape perception scores under epidemic conditions, while vaccinated individuals were more tolerant of various noises. Recommendations for landscape design to improve the soundscape perception of elderly people are accordingly provided.

**Keywords:** soundscape preference; elderly; urban forest park; underdeveloped cities in China; subjective evaluation

## 1. Introduction

As an important ecological service system for residential areas, urban forest parks are often referred to as the "heart of the city" due to their important role in maintaining the

overall urban environment [1,2]. They harbor all urban trees, shrubs, lawns, pervious soils, roads, and other landscape facilities [3]. A quality urban forest park may enhance health, cleanse the air, and offer space for ecotourism, entertainment, and exercise, as well as aid in the prevention of obesity and the alleviation of chronic illnesses [4,5]. By concentrating on the relationships between individuals, sound perception, the acoustic environment, and society, soundscape studies regard the acoustic environment as a resource rather than as noise [6]. In addition, studies have shown that urban forest parks play an active role in providing mental relaxation via acoustic landscape design and implementation [7,8] Soundscape perception can be employed in the design and maintenance of forest parks to make them more appealing to park visitors. This not only serves to maximize visitor–park interactions but also helps to improve visitors' experience of the parks [9]. Recent years have witnessed a global concern for mental and physical well-being. Interestingly, research has shown that the proportion of people with depression is lowest near the equator and highest near the poles. Vulnerable groups, such as the elderly and patients with chronic illnesses, generally have a poorer understanding of weather-related risks [10], which further affects their mental and physical health [11]. Therefore, studying the activities of the elderly in high-latitude regions can contribute to understanding how to improve their well-being.

Meanwhile, urban forest parks are becoming increasingly significant to middle-aged and older people as the population continues to age and as their significance in residents' routines increases [9,12]. Urban forest parks are distinct from both urban parks and forest parks in that they are typically found in suburban or central metropolitan areas. The forest biological environment serves as a support system, while the human-made natural landscape serves as a complement [13]. Some of the benefits of urban forest parks are their outstanding acoustic environments, which can help with the elderly population's health issues.

In underdeveloped cities, the connection between the elderly and urban forest parks is even more prominent. Herein, underdeveloped cities refer to regions that have some economic strengths and potentialities but still lag behind developed regions, with uneven productivity development and underdeveloped technologies. Typical cases are the central, western, and northeastern regions of China [14]. Irregularities are not uncommon in the distribution of age groups in such residential areas. With younger age groups flooding into large, advanced, or prosperous developed metropolitan areas, the proportions of older age groups in less-developed urban areas are increasing to new highs. Due to a lack of recreational choices in these areas, parks are some of the primary locations for the aged groups to visit. It has been found that in these areas, older people spend 62.43% of their daytime hours in parks [15]. After the COVID-19 pandemic, a further decrease was observed in indoor activities among elder groups, and a further increase in the significance of urban forest parks was observed.

Compared to younger individuals, elderly people have lower levels of communication with the outside world and are less able to provide feedback in response to their surroundings [16]. According to demographic census data, the percentage of the population aged 60 and older nationwide had reached 18.7% by 2020, with the percentage of people aged 65 and older amounting to 13.5%. The aging problem is particularly salient in the Northeast, Sichuan, and Chongqing regions, with each surpassing 20% by 2020. Notably, these are also the least-developed regions in China [14]. Older adults may be more susceptible to experiencing higher levels of HA, which can significantly lower their quality of life and social adaptability. This is particularly true for countries such as Bulgaria, where there has been a steady increase in the proportion of older people at risk of poverty over the past decade [15].

In other words, it is impossible to overestimate the significance of the forest park as an area firmly connected to the elderly. The natural conditions in forest parks might vary, which may have an impact on the well-being of elderly individuals [17]. Soundscape factors in the natural environment in particular can have a direct impact on the mental health, cognitive function, and physical functioning of older adults [18]. Studies have

revealed that elder adults are prone to many physical ailments, such as chronic obstructive pulmonary disease and cognitive decline [19], for which soundscape factors are crucial for an elderly person's physical recovery [20].

The word "soundscape" is used in the interdisciplinary field known as "soundscape studies," which was established by activist and composer R. Murray Schafer [21]. In the same way that visual pictures displayed in a specific location are considered landscapes in general, soundscapes may be thought of as audio landscapes [6]. Several studies on urban park soundscapes have been performed by academics in Asia, Oceania, Europe, and America [22], and it has been discovered that soundscape elements have an impact on the physical and mental health of children, the elderly, and those who are blind [23,24]. According to the study Soundscape Preference of Urban Peoples in China in the Post-Pandemic Era, soundscape preference is the preference of one or more persons for the sound environment of an area. Existing research shows that the elderly may suffer from a variety of medical conditions, including chronic obstructive pulmonary disease and cognitive decline [25,26], on which environmental soundscape elements can have a significant impact [27,28]. Additionally, urban forest parks can benefit elderly individuals' physical health and offer soundscapes [29,30]. It has been found that as society evolves, parks have become hideaways for older adults to escape the "urban disease" [31]. In the aftermath of the COVID-19 pandemic, the use of masks has increased greatly among the public. The elderly, with their organs becoming fragile, have a strong demand for soundscapes which is even more intense as wearing a mask greatly decreases the perceived level of a soundscape [32,33].

Nevertheless, in recent years, fewer researchers have concentrated on the findings of studies on audiovisual aspects conducted by Japanese researchers who combined forest-derived audiovisual stimuli to induce physiological and psychological relaxation, with the physiological relaxation effect being more pronounced under such conditions [34]. It is worth noting that no study has been carried out in this crucial research area on how the elderly perceive such environments in China's underdeveloped towns. With the advancement of society and technology, a variety of methods have emerged for surveying large numbers of respondents. Meanwhile, traditional data collection techniques, such as the subjective evaluation method and the questionnaire method, have been downplayed because of their inefficiencies in gathering big data [35].

Due to the limited knowledge of the elderly, the majority of previous studies on elderly populations have employed the more traditional subjective evaluation approach in combination with questionnaires to ensure that older adults can be maximally engaged in the studies. In addition, despite the heavy burden, the interaction with the elderly enables researchers to gain a deeper level of comprehension of the issue under examination [36].

The main objective of this study, which used Maoershan Forest Park as an example, was to investigate which acoustic elements of the park have an impact on the cognition of elderly individuals and their preferences for acoustic environments, as well as to provide suggestions for improving the mental and physical well-being of the elderly. A questionnaire was designed and administered to survey the preferred soundscape of the elderly in the underdeveloped city. A statistical analysis was then performed to detect the variables that affect soundscape preference. Finally, a regression coefficient model was developed to investigate the elderly participants' preferences for soundscapes. Answering the questions above helped us understand the soundscape preferences of elderly people in urban forest parks in China's underdeveloped urban areas, and our research team also offers some advice on how to create a soundscape.

## 2. Methodology

This research was based on the important theoretical foundations of ecology, acoustics, and landscape architecture, as well as the concept and academic background of urban public space soundscapes. The preferences of elderly people in urban forest parks in underdeveloped cities in China were investigated using a subjective evaluation method.

Specifically, a survey questionnaire was designed and administered to a group of frequent forest park visitors in the underdeveloped city of Yanji in Northeast China.

The methodological design of this study is demonstrated as follows (Figure 1).

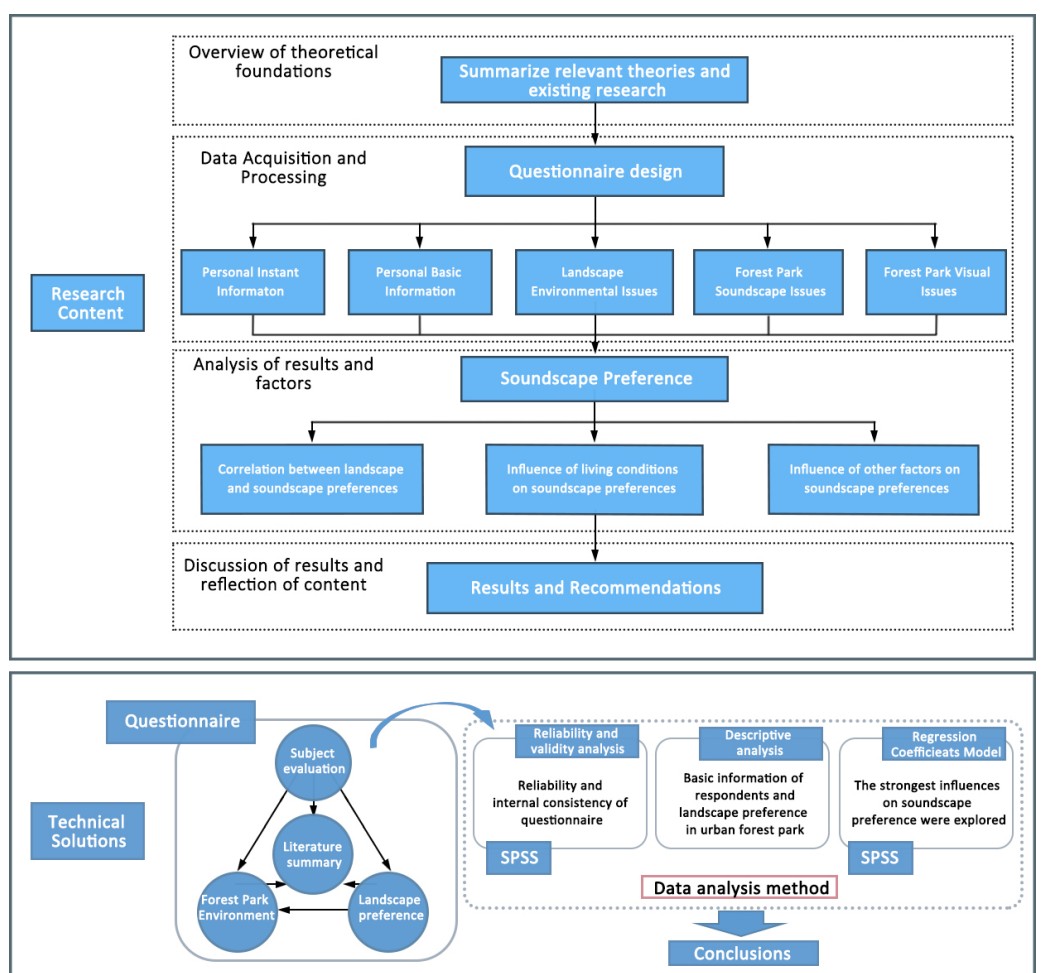

**Figure 1.** Research Contents and technical methods.

*2.1. Research Context*

Underdeveloped regions, due to limited healthcare conditions, inadequate economic development, and the mobility of younger populations, often have relatively high proportions of elderly populations. Research has shown that natural environments, particularly forest parks, have positive impacts on the physical and mental health of the elderly. Forest parks offer good air quality and atmospheres that are conducive to alleviating stress and anxiety among the elderly, promoting physical activity and social engagement, and enhancing happiness and quality of life. Therefore, studying the needs and utilization of forest parks among the elderly in underdeveloped regions can provide a better understanding of this specific population's demands and health conditions. In underdeveloped regions where healthcare resources may be limited, natural environments can become vital resources for the elderly to maintain their health, and studying the health benefits of forest parks among the elderly in such regions can provide scientific evidence for improving their quality of life.

China's population is the largest population in the world, and its elderly population is also the largest. Despite China's fast overall economic growth in recent decades, about 80% of the provinces and cities in China are still underdeveloped, with disproportionately large elderly populations and inefficient supplies of medical services. The city of Yanji in Jilin Province is a typical case in point. It has been rated as one of the key cities in northeast

China since China's reform and opening up. However, it is still in an underdeveloped state and faces many problems: there are fewer large projects, the supporting industries still rely on traditional industries, and the industrial structure is not balanced. Additionally, the city's transformation and upgrading still have a long way to go; the image of the capital city has not been fully manifested, the supporting functions of the city still need to be improved, the degree of refinement of urban management needs to be strengthened, the level of grass-roots social governance needs to be improved, and there is still much room for improving the quality of the city. Finally, the pressure on ecological environmental protection is extremely strong [37,38]. It is more worthwhile to investigate cities such as Yanji than provincial capitals such as Harbin, which is developing slowly and yet has a large population [7,39,40].

Since the central government's call to revive the northeast, Yanji, as a city near China's border with North Korea, is now facing the pressure of finding a clear path in its future development, partially due to its prolonged history of underdevelopment [41]. Realistic strategies are urgently needed to build up infrastructure that can ensure proper and efficient growth [42]. According to the data from the sixth population census, the proportion of the elderly population in Yanji City is approximately 22% [43]. The city's chronic underdevelopment has resulted in a large decline in the proportion of young people and an increase in the number of senior residents, making the problem of retirement all the more difficult for the local government authorities [44].

In spite of its underdevelopment, Yanji hosts several national forest parks, of which the Maoershan National Forest Park (its location is shown in Figure 2) is of significant value to the current research study as it is the primary urban forest park in Yanji and one of the largest urban forest parks in the neighboring provinces and cities. As noted above, forest parks can contribute to community development and economic growth. Underdeveloped regions often face economic challenges and social development issues. Studying the utilization of forest parks among the elderly in underdeveloped regions can help governments and community organizations understand the needs of the elderly, formulate relevant policies and plans, and promote community development and elderly participation. To put it simply, choosing the Maoershan National Forest Park in Yanji, Jilin, China, for the study of forest parks among the elderly in underdeveloped regions is not only helpful in gaining deeper insights into the needs and health conditions of this specific population but also in garnering scientific evidence for improving their quality of life and promoting community development.

In addition, the Maoershan National Forest Park has a natural environment that fits the purpose of the current study in that it has the characteristics of mountains, water, forests, fields, and cities and reflects the customs of Korean people. The park is home to a variety of pine trees, elm trees, poplar trees, and shrubs, wild animals such as pheasants and hares, and mushrooms. The climate is a temperate monsoon climate, with a dry and windy spring, a rainy season from June to August, and a cool autumn and cold winter. The average annual temperature is 2 °C–6 °C, with extreme minimum temperatures of −23 °C–34 °C and maximum temperatures of 34 °C–38 °C. The annual sunshine hours range from 2150–2480 h, and the average annual precipitation is 400 mm–650 mm. In addition, the Maoershan National Forest Park is easily accessible, most areas are free of charge, and the park has excellent forest ecological landscape resources and superior conditions for developing forest tourism [45]. The average daily visitor count at the forest park is approximately 10,000, and during holidays, it can reach a maximum of 60,000. The considerable number of visitors per day makes it possible to ensure ecological diversity for the current study.

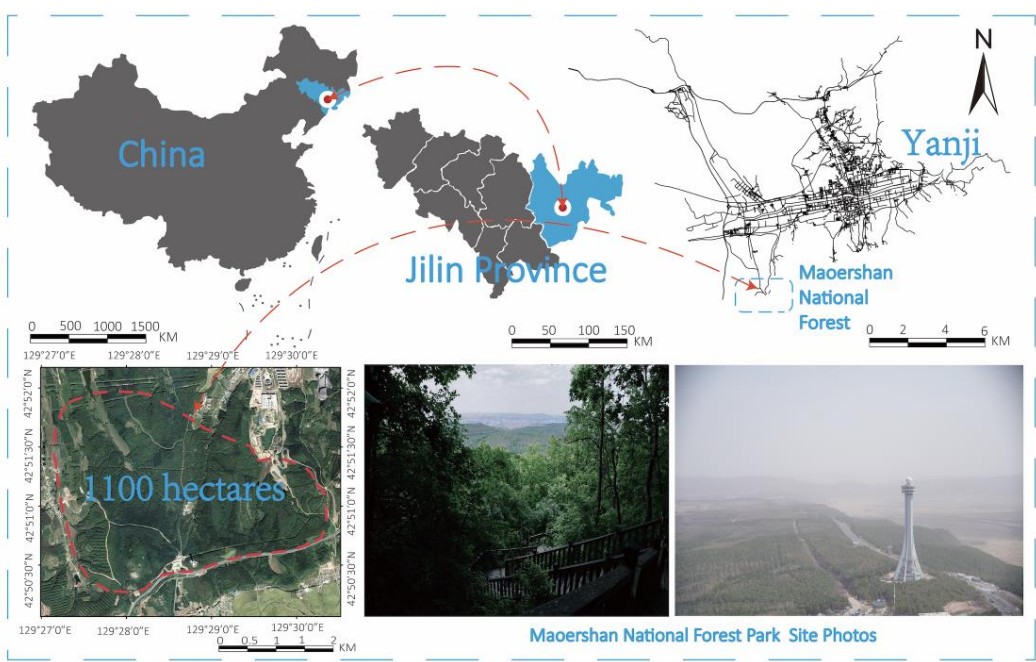

**Figure 2.** Location of the Maoershan National Forest Park in Yanji City.

Moreover, the Maoershan National Forest Park is geographically close to our unit, and our team was able to obtain information about what the elderly population cares about in the area and feedback from local residents regarding their daily lives. It can thus be used as an ideal example to explain the preferred soundscapes for elderly people living in undeveloped cities and to offer suggestions for the landscape planning and design of the forest park.

## 2.2. Research Design

The respondents were chosen at random from residential districts close to Maoershan National Forest Park in Yanji City to guarantee the questionnaire's thoroughness and to guarantee that the respondents were representative of the research. A pilot survey was first conducted in January 2022 in which the research team employed the simple random sampling method to select 68 elderly visitors to Maoershan National Forest Park. After cleaning the collected data, a total of 57 valid questionnaires were obtained, with a valid response rate of 83.8%. Among the 57 respondents, 25 were aged 60 and above, accounting for 30.2% of the total. The questionnaire was then adapted to avoid any misunderstanding among the elderly participants, and before conducting the formal survey, the surveyors received additional training that was aimed to facilitate the research process for all elderly individuals with normal hearing and to minimize the potential for misinterpretation. To identify survey participants, we randomly sampled adults who were able to subjectively assess public landscapes and soundscapes. Before the completion of the survey, a quick hearing test was performed to make sure that all chosen older adults had a normal degree of hearing. All respondents found to have a hearing problem or who were unable to understand the surveyor's instructions were disqualified.

The field investigation spanned from 10 February 2022 to 1 March 2022, and we visited the site for the survey on eighteen days with sunny weather within that stretch of time. Five volunteers were recruited to assist with the survey work. The combination of soundscapes with the routes and activity areas are displayed in Figure 3.

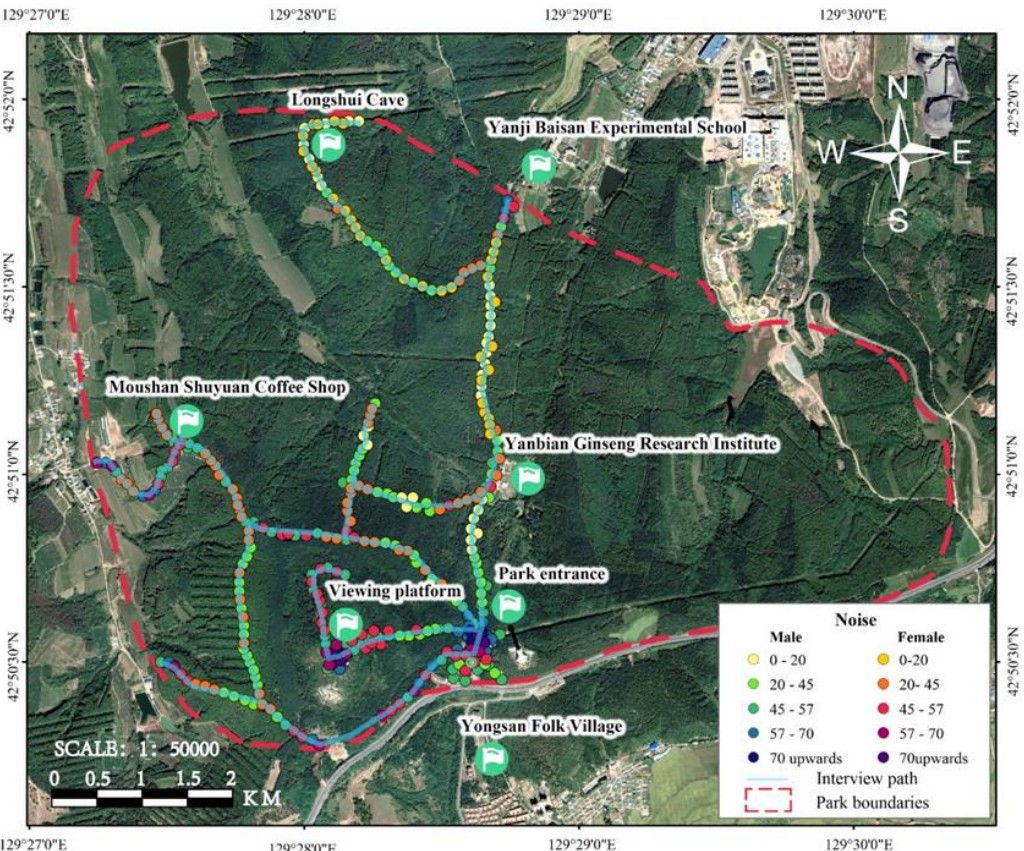

**Figure 3.** Research route map with surrounding sound environments.

To guarantee that the questionnaire would not take too long to complete, a pre-study was completed two days ahead of the survey. A small present was prepared to encourage the locals to participate in the survey. During the survey, a certain number of respondents were selected at random at several traffic intersections close to the park between 7 am and 10 am, between 3 pm and 6 pm, and between 7 pm and 9 pm. The total number of valid subjects was 725, accounting for approximately 10% of the daily visitor traffic. The respondents' key demographic information is displayed in Table 1.

The respondents were asked to complete the questionnaire by using the tablet PCs provided to them by the researchers. The questionnaire was delivered by "Questionnaire Star", a professional and authoritative questionnaire production and distribution platform which has published 154 million questionnaires and could completely fulfill the demands of this survey, including the quantity of questionnaires and question types. Special investigators were arranged to help the respondents with the questionnaire completion process. The researchers also prepared a moderate number of paper-based questionnaires for senior people who were unable to utilize electronic devices properly.

The questionnaire used in this study was designed in accordance with the existing literature and in keen consideration of the characteristics. The questionnaires used in previous related research studies were usually comprised of two general sections: a subjective section that aimed to collect the participants' personal responses toward the items under investigation and an objective section that pertained to environmental information, among other things [46]. It has been found that individual preferences are also influenced by immediate factors [4,47] such as visual features, especially the intricate coupling of auditory and visual elements [48]. In light of these findings, the questionnaire was designed to comprise 46 questions in total that were divided into five sections: time information, basic personal information, landscape environment, soundscape key, and initiatory environment.

**Table 1.** Demographic account of respondents.

| | Classification | Percentage |
|---|---|---|
| Gender | Female | 48.70% |
| | Male | 51.30% |
| Age | 60–64 | 43.30% |
| | 65–69 | 34.90% |
| | 70–74 | 12.00% |
| | 75–79 | 9.70% |
| | 80–84 | 0.10% |
| Education | Junior High School and Below | 48.00% |
| | High School or Vocational school | 42.90% |
| | Junior College or Undergraduate | 9.00% |
| | Master's degree and Above | 0.10% |
| Pension | EUR 0–100 | 26.10% |
| | EUR 101–300 | 73.70% |
| | EUR > 300 | 0.10% |
| Occupation | Experts, technicians and related workers | 4.40% |
| | Government officials and business managers | 4.60% |
| | Sales professionals | 17.00% |
| | Service professionals | 24.80% |
| | Agricultural, animal husbandry and forestry workers, fishermen and hunters | 20.40% |
| | Manufacturers and production-related workers, transportation equipment operators and workers | 18.80% |
| | Workers who cannot be classified by occupation | 10.10% |
| Physical Condition | Completely self-reliant | 89.80% |
| | In need of care | 10.20% |

The first section aimed to gather immediate data from the respondents as it has been found that the respondents' perceptions of the soundscape can be influenced to varying degrees by some external factors, such as the weather, i.e., how the climate felt that day, how they felt when they visited the forest that day, and how they felt on that day [49,50].

The second section focused on the respondents' personal information, including gender, age, education, occupation, pension, physical condition, visual condition, auditory condition, residence, the length of time it took to reach the forest park from where they lived, how often they visited the forest park, how they arrived at the forest park, why they went to the forest park, when they arrived at the forest park, how long they were inclined to stay at the forest park, what kind of signage they valued most in the forest park, whether the forest park signs could be obviously felt, the function of the forest park buildings, what kind of environment they liked best when staying in the forest park, and which buildings and services should be more important in the forest park [51,52].

The third section pertained to the question of landscape issues, specifically, the kind of water body that was preferred, the kind of tree environment that was preferred, whether pure green or color accents were preferred, and what kind of sky was preferred, as well as the visitors' understanding of noise, i.e., whether noise had an impact on their experience, whether they had complaints about noise, how to deal with noise pollution, how to reduce noise, and how to hear the location of the noise [53].

The fourth section aimed to collect information related to forest park soundscape issues. Respondents were expected to indicate the category and rate the volume of the vehicle sound, the bird song, the musical sound, the natural sound, and other sound in the forest park [24]. The last section concerned the visitors' perceptions of the visual aspects of the forest park, including their visual perceptions of the booths and icons established in the park. A 5-point Likert scale (strongly dislike (−2), dislike (−1), average (0), like (1), and like very much (2)) was adopted in the fourth and fifth sections to indicate their overall soundscape preferences. The pre-study administration of the questionnaire with 20 people

showed that the average response time was 5 min and 19 s and that the questions were well designed as no respondents reported any doubts or objections.

*2.3. Reliability and Validity Assessment*

2.3.1. Reliability Analysis

First, the Cronbach coefficient method was used to test the internal consistency of each dimension. The Cronbach coefficient takes values in the range of 0–1, and the higher the value of the coefficient, the better the reliability. In general, a coefficient of confidence below 0.6 is not credible, between 0.6 and 0.7 is credible, between 0.7 and 0.8 is relatively credible, between 0.8 and 0.9 is very credible, and between 0.9 and 1 is very credible. Based on the analysis results shown in Table 2, the reliability coefficient of vehicle sound in this analysis was 0.938, which fell within the range of 0.9–1, indicating that the vehicle sound dimension had a very good internal consistency and a very good reliability. The reliability coefficient of musical sound, which had the lowest value, is 0.870, within the range of 0.8–0.9, indicating that the reliability of the musical sound dimension was very credible. Taken together, all variables exhibited good reliability.

**Table 2.** Results of the reliability analysis of each variable.

| Variable | Abbreviations | Cronbach Alpha | Number of Items |
|---|---|---|---|
| Vehicle Sound | VS | 0.938 | 12 |
| Bird Song | BS | 0.928 | 8 |
| Livestock Sound | LS | 0.938 | 9 |
| Atmospheric Sound | AS | 0.885 | 5 |
| Musical Sound | MS | 0.870 | 3 |
| Natural Sound | NS | 0.874 | 4 |
| Other Sound | OS | 0.934 | 9 |
| Vision of Park Stands | VPS | 0.885 | 6 |
| Vision of Other Things | VOT | 0.872 | 6 |
| Vision of Signs | VOS | 0.834 | 2 |

2.3.2. Validity Analysis

An exploratory factor analysis was used to test the structural validity of the study. In this study, there were 10 pre-defined dimensions in the scale section, and each dimension contained a certain number of measurement items. The results of the final factor categorization of the component matrix or the rotated component matrix were observed via exploratory factor analysis. If the categorization results are consistent with the predefined dimensions, the scale has good structural validity.

KMO values range from 0 to 1. The higher the coefficient value, the more suitable the data are for factor analysis. Generally, if the KMO value is less than 0.6, it is not suitable for factor analysis. As shown in Table 3, the KMO value was 0.922, which means that the dataset was suitable for factor analysis. In addition, the Bartlett test result ($p < 0.001$) rejected the original hypothesis, indicating that the data collected in this study were very suitable for factor analysis.

**Table 3.** KMO and Bartlett test.

| **KMO Values** | | **0.922** |
|---|---|---|
| | Approximate cardinality | 30,920.235 |
| Bartlett test | Degree of freedom | 2016 |
| | Significance | <0.001 |

### 2.3.3. Component Matrix after Transposition

The maximum variance method was used to extract the principal components according to the criteria of eigenvalues greater than 1. Finally, a total of 10 principal components were extracted, and the cumulative variance contribution rate was 67.02% indicating that the principal components extracted in this analysis could effectively replace the original data set (Table 4). The results of the commonality analysis showed that the commonality of each question item reached a standard of 0.4 or more, indicating that the results of categorizing the 10 principal component items were consistent with the preset dimensions of the questionnaire. Additionally, the factor loadings of each question were all greater than 0.5, basically above 0.7.

**Table 4.** Component matrix after transposition.

| | 1 | 2 | 3 | 4 | 5 | 6 | 7 | 8 | 9 | 10 |
|---|---|---|---|---|---|---|---|---|---|---|
| | VS1 | LS1 | OS1 | BS1 | VPS1 | VOT1 | AS1 | NS1 | MS1 | VOS1 |
| | VS2 | LS2 | OS2 | BS2 | VPS2 | VOT2 | AS2 | NS2 | MS2 | VOS2 |
| | VS3 | LS3 | OS3 | BS3 | VPS3 | VOT3 | AS3 | NS3 | MS3 | |
| | VS4 | LS4 | OS4 | BS4 | VPS4 | VOT4 | AS4 | NS4 | | |
| | VS5 | LS5 | OS5 | BS5 | VPS5 | VOT5 | AS5 | | | |
| | VS6 | LS6 | OS6 | BS6 | VPS6 | VOT6 | | | | |
| Measure question items | VS7 | LS7 | OS7 | BS7 | | | | | | |
| | VS8 | LS8 | OS8 | BS8 | | | | | | |
| | VS9 | LS9 | OS9 | | | | | | | |
| | VS10 | | | | | | | | | |
| | VS11 | | | | | | | | | |
| | VS12 | | | | | | | | | |
| Cumulative variance contribution rate | | | | | 67.02% | | | | | |

All abbreviations are shown in Table 2.

## 3. Results

### 3.1. Descriptive Statistics of the Elders' Preferences for Forest Park Soundscape

It can be seen in Table 5, that the elderly participants liked the sound made by nature in the forest park the most (mean = 3.66), followed by livestock sound (mean = 3.54), musical sound (mean = 3.45), and sound from birds of prey (mean = 3.50), and they disliked vehicle sound (mean = 2.35), atmospheric sound (mean = 2.34), and other sound (mean = 2.48) as well. In the category of natural sound, they liked the sound of rustling leaves the most (mean = 3.78) and the sound of falling stones the least (mean = 3.60); in the category of livestock sound, they liked the sound of cows the most (mean = 3.64) and the sound of goose the least (mean = 3.46); in the category of musical sound, they liked the sound of musical instruments the most (mean = 3.50) and the sound of electronic technology the least (mean = 3.50). In the category of musical sound, they liked the sound of musical instruments the most (mean = 3.50) and the sound of electronic technology products the least (mean = 3.42).

**Table 5.** Average description of the elders' overall soundscape preferences.

| Sound Category | Code | Sound Source | Average | Total Average |
|---|---|---|---|---|
| Vehicle Sound | 1 | Car | 2.22 | |
| | 2 | Bus | 2.34 | |
| | 3 | Express train | 2.35 | |
| | 4 | Aircraft | 2.36 | |
| | 5 | Fighter | 2.34 | |
| | 6 | Motorcycle | 2.35 | 2.35 |
| | 7 | Tractor | 2.36 | |
| | 8 | Bicycle | 2.38 | |
| | 9 | Truck | 2.34 | |
| | 10 | Police siren | 2.35 | |
| | 11 | Ambulance siren | 2.37 | |
| | 12 | Fire engine siren | 2.39 | |
| Bird Song | 13 | Pigeon | 3.61 | |
| | 14 | Wild goose | 3.55 | |
| | 15 | Swallow | 3.48 | |
| | 16 | Eagle | 3.49 | |
| | 17 | Hawk | 3.49 | 3.50 |
| | 18 | Swan | 3.50 | |
| | 19 | Egret | 3.45 | |
| | 20 | Sparrow | 3.46 | |
| Livestock Sound | 21 | Cattle | 3.64 | |
| | 22 | Horse | 3.50 | |
| | 23 | Sheep/Goat | 3.50 | |
| | 24 | Chicken | 3.54 | |
| | 25 | Dog | 3.52 | 3.54 |
| | 26 | Pig | 3.54 | |
| | 27 | Duck | 3.58 | |
| | 28 | Cat | 3.54 | |
| | 29 | Goose | 3.46 | |
| Atmospheric Sound | 30 | Rain | 2.24 | |
| | 31 | Wind | 2.35 | |
| | 32 | Snow | 2.33 | 2.34 |
| | 33 | Thunder | 2.38 | |
| | 34 | thunderstorm | 2.38 | |
| Musical Sound | 35 | Instrumental | 3.50 | |
| | 36 | Vocal | 3.43 | 3.45 |
| | 37 | Electronic | 3.42 | |
| Natural Sound | 38 | Leaves | 3.78 | |
| | 39 | Falling stone | 3.60 | 3.66 |
| | 40 | Flying dust | 3.61 | |
| | 41 | Flowing water | 3.64 | |
| Other Sound | 42 | Mechanical | 2.36 | |
| | 43 | Construction site | 2.48 | |
| | 44 | Handwork | 2.51 | |
| | 45 | Human activities | 2.48 | |
| | 46 | Mobile ringtones | 2.50 | 2.48 |
| | 47 | Children playing | 2.44 | |
| | 48 | Street performance | 2.54 | |
| | 49 | Sneezing | 2.51 | |
| | 50 | Nonlocal dialect | 2.52 | |

As shown in Figure 4, the specific sound can have particularly negative effects on elders' psychological perceptions.

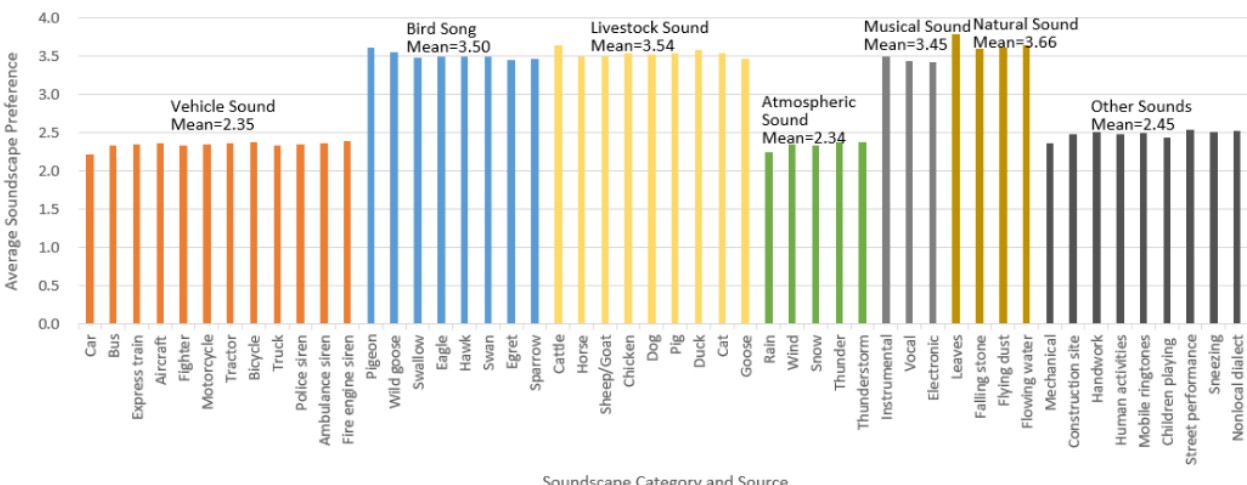

**Figure 4.** Columnar distribution of the mean value of soundscape preference (code is shown in Table 1).

### 3.2. Correlation between Landscape and Soundscape Preference

According to the correlation analysis (Table 6), the type of public landscape space and landscape structural design both influenced the subjective evaluation of the soundscape. In terms of overall landscape perception, those who preferred areas with sunshine demonstrated relatively higher tolerances for the sound of cars, buses, bicycles, fire trucks, ambulance sirens, apparatus, construction, handicrafts, human activities, stranger calls, children playing, street vendors, and local dialects. Those who preferred environments under the shade of trees had higher degrees of preference for the sound of planes, motorcycles, ambulance sirens, fire truck sirens, rainstorms, freezing rain, thunder, sound of instruments, children playing, etc. Those who preferred waterside environments had higher degrees of preference for the sound of passenger cars, ambulance sirens, heavy rain, wind, rain, and lightning. In the selection of landscape structural design, those who used walkways had relatively low preferences for the sound of police car sirens, while respondents who took the bus had relatively low preferences for the sound of leaves. Seniors who used private cars had relatively low preferences for the sound of the creek trickling. Elderly people who rode bicycles to the forest park had relatively low preferences for the sound of goose.

### 3.3. Influence of Living Conditions on Soundscape Preference

The results of the correlation analysis (Table 7) showed that the distance (Q12) to the green space, the frequency (Q13) of visiting the forest park, and the purpose (Q15) were related to the degree of soundscape preference. It can be inferred from the correlation results that (1) in terms of distance, the closer the elder was to the forest park, the higher their tolerance for sounds of birds, livestock, music, and nature; (2) in terms of the frequency of visiting the forest park, the higher the frequency of visiting green space, the more the elders liked the sounds of birds, livestock, music, and nature; (3) the seniors who visited the forest park for exercise preferred livestock sound, musical sound, and natural sound; (4) seniors who visited the Forest Park to walk their dogs and play chess preferred vehicle sound, atmospheric sound, and other sound; and (5) seniors who visited the forest park for square dancing preferred vehicle sound and atmospheric sound.

**Table 6.** Correlation between public landscape space preference and soundscape preference.

| Sound Category | Code | Sound Source | Q21: Which Environment Do You Prefer to Stay in the Forest Park? | | | | | | | | | | | Q14: How Do You Get to the Forest Park Now? | | | |
|---|---|---|---|---|---|---|---|---|---|---|---|---|---|---|---|---|---|
| | | | In Sunlit Areas | In the Shade of Trees | Near a Snack Stall | In Secluded Areas | On the Water-side | In Lounge Areas | In Open Pave-ments | On Roadside Seats | In Higher Places | In Park Build-ings | On the Open Lawn | On Foot | By Bus | By Car | By Bike |
| Vehicle Sound | 1 | Car | 0.100 ** | 0.110 ** | 0.072 | 0.134 ** | 0.064 | −0.061 | −0.003 | −0.073 | 0.071 | −0.004 | −0.008 | −0.061 | 0.080 * | 0.114 ** | 0.015 |
| | 2 | Bus | 0.087 * | 0.029 | 0.038 | 0.102 ** | 0.083 * | −0.011 | 0.018 | −0.038 | 0.102 ** | −0.001 | −0.026 | −0.055 | 0.058 | 0.129 ** | 0.004 |
| | 3 | Express train | 0.057 | 0.072 | 0.060 | 0.059 | 0.012 | −0.021 | −0.033 | −0.044 | 0.040 | −0.026 | 0.002 | −0.055 | 0.024 | 0.066 | 0.028 |
| | 4 | Aircraft | 0.053 | 0.088 * | 0.047 | 0.110 ** | 0.027 | −0.034 | 0.021 | −0.059 | 0.026 | −0.002 | −0.030 | −0.026 | 0.041 | 0.038 | 0.043 |
| | 5 | Fighter | 0.038 | 0.020 | 0.008 | 0.079 * | 0.046 | −0.076 * | −0.004 | −0.061 | 0.025 | 0.050 | 0.081 * | −0.009 | 0.030 | 0.060 | 0.011 |
| | 6 | Motorcycle | 0.064 | 0.092 * | 0.077 * | 0.049 | 0.033 | −0.041 | 0.021 | −0.052 | 0.040 | −0.014 | −0.026 | −0.041 | 0.052 | 0.085 * | −0.001 |
| | 7 | Tractor | 0.046 | 0.068 | 0.045 | 0.128 ** | 0.054 | −0.078 * | 0.012 | −0.041 | 0.040 | 0.024 | 0.023 | −0.005 | 0.030 | 0.086 * | 0.035 |
| | 8 | Bicycle | 0.086 * | 0.066 | 0.000 | 0.084 * | 0.016 | −0.023 | 0.053 | −0.019 | 0.023 | −0.016 | −0.011 | −0.033 | 0.042 | 0.064 | 0.055 |
| | 9 | Truck | 0.057 | −0.023 | 0.006 | 0.020 | 0.031 | 0.025 | 0.031 | −0.001 | 0.010 | −0.013 | −0.045 | 0.015 | 0.047 | 0.018 | 0.042 |
| | 10 | Police siren | 0.069 | 0.087 * | 0.067 | 0.125 ** | 0.044 | −0.039 | −0.032 | −0.074 * | 0.072 | −0.027 | 0.031 | −0.076 * | 0.075 * | 0.077 * | 0.045 |
| | 11 | Ambulance siren | 0.113 ** | 0.122 ** | 0.071 | 0.124 ** | 0.076 * | −0.098 ** | −0.046 | −0.126 ** | 0.073 | −0.003 | 0.007 | −0.075 * | 0.082 * | 0.115 ** | 0.036 |
| | 12 | Fire engine siren | 0.091 * | 0.132 ** | 0.097 ** | 0.114 ** | 0.021 | −0.027 | 0.010 | −0.100 ** | 0.022 | 0.034 | −0.036 | −0.071 | 0.051 | 0.110 ** | 0.049 |
| Bird Song | 13 | Pigeon | −0.118 ** | −0.191 ** | −0.191 ** | −0.207 ** | −0.178 ** | 0.127 ** | 0.007 | 0.133 ** | −0.037 | −0.057 | 0.081 * | 0.094 * | −0.104 ** | −0.099 ** | −0.099 ** |
| | 14 | Wild goose | −0.124 ** | −0.186 ** | −0.176 ** | −0.158 ** | −0.158 ** | 0.127 ** | −0.028 | 0.091 * | −0.017 | −0.050 | 0.064 | 0.115 ** | −0.090 * | −0.114 ** | −0.080 * |
| | 15 | Swallow | −0.141 ** | −0.140 ** | −0.157 ** | −0.152 ** | −0.146 ** | 0.145 ** | 0.024 | 0.113 ** | −0.027 | −0.068 | 0.077 * | 0.090 * | −0.066 | −0.088 * | −0.096 ** |
| | 16 | Eagle | −0.121 ** | −0.127 ** | −0.133 ** | −0.112 ** | −0.098 ** | −0.098 ** | 0.029 | 0.081 * | −0.042 | −0.046 | 0.039 | 0.102 ** | −0.094 * | −0.079 * | −0.064 |
| | 17 | Hawk | −0.115 ** | −0.151 ** | −0.174 ** | −0.155 ** | −0.194 ** | 0.102 ** | 0.013 | 0.109 ** | −0.013 | −0.022 | 0.104 ** | 0.104 ** | −0.097 ** | −0.069 | −0.116 ** |
| | 18 | Swan | −0.167 ** | −0.166 ** | −0.112 ** | −0.141 ** | −0.116 ** | 0.092 * | 0.022 | 0.131 ** | 0.000 | −0.083 * | 0.073 * | 0.085 * | −0.058 | −0.068 | −0.093 * |
| | 19 | Egret | −0.092 * | −0.143 ** | −0.152 ** | −0.128 ** | −0.115 ** | 0.110 ** | 0.022 | 0.081 * | −0.011 | −0.043 | 0.017 | 0.047 | −0.103 ** | −0.044 | −0.060 |
| | 20 | Sparrow | −0.104 ** | −0.200 ** | −0.169 ** | −0.146 ** | −0.145 ** | 0.083 * | −0.039 | 0.115 ** | 0.002 | −0.068 | 0.088 * | 0.104 ** | −0.109 ** | −0.087 * | −0.082 * |
| Livestock Sound | 21 | Cattle | −0.075 * | −0.071 | −0.148 ** | −0.163 ** | −0.107 ** | 0.113 ** | 0.008 | 0.124 ** | −0.024 | −0.011 | 0.013 | 0.057 | −0.039 | −0.148 ** | −0.100 ** |
| | 22 | Horse | −0.058 | −0.057 | −0.086 * | −0.128 ** | −0.141 ** | 0.103 ** | 0.021 | 0.096 ** | 0.000 | −0.024 | 0.016 | 0.070 | −0.017 | −0.118 ** | −0.079 * |
| | 23 | Sheep/Goat | −0.059 | −0.071 | −0.104 ** | −0.133 ** | −0.111 ** | 0.074 * | −0.006 | 0.145 ** | −0.087 * | −0.023 | 0.053 | 0.060 | −0.060 | −0.091 * | −0.072 |
| | 24 | Chicken | −0.062 | −0.048 | −0.102 ** | −0.194 ** | −0.124 ** | 0.101 ** | 0.052 | 0.156 ** | −0.017 | −0.039 | 0.001 | 0.062 | −0.037 | −0.102 ** | −0.108 ** |
| | 25 | Dog | −0.030 | −0.103 ** | −0.111 ** | −0.147 ** | −0.096 ** | 0.110 ** | 0.010 | 0.100 ** | −0.030 | −0.037 | 0.009 | 0.040 | −0.057 | −0.064 | −0.116 ** |
| | 26 | Pig | −0.069 | −0.044 | −0.065 | −0.136 ** | −0.108 ** | 0.080 * | 0.011 | 0.110 ** | 0.012 | −0.003 | −0.004 | 0.027 | 0.002 | −0.094 * | −0.049 |
| | 27 | Duck | −0.063 | −0.064 | −0.144 ** | −0.137 ** | −0.126 ** | 0.070 | 0.032 | 0.057 | −0.019 | −0.041 | 0.028 | 0.067 | −0.056 | −0.114 ** | −0.098 ** |
| | 28 | Cat | −0.008 | −0.064 | −0.115 ** | −0.175 ** | −0.121 ** | 0.056 | 0.003 | 0.091 * | −0.017 | 0.009 | 0.039 | 0.024 | −0.037 | −0.104 ** | −0.084 * |
| | 29 | Goose | −0.050 | −0.103 ** | −0.087 * | −0.151 ** | −0.079 * | 0.064 | 0.006 | 0.087 * | −0.041 | −0.057 | −0.002 | 0.057 | 0.007 | −0.104 ** | −0.137 ** |
| Atmospheric Sound | 30 | Rain | 0.089 * | 0.084 * | 0.100 ** | 0.108 ** | 0.090 * | −0.044 | −0.019 | −0.073 * | 0.035 | 0.093 * | −0.027 | −0.022 | 0.035 | 0.054 | 0.085 * |
| | 31 | Wind | 0.071 | 0.071 | 0.073 * | 0.070 | 0.068 | −0.037 | 0.036 | −0.063 | 0.010 | 0.038 | 0.001 | 0.001 | 0.026 | 0.034 | 0.060 |
| | 32 | Snow | 0.058 | 0.083 * | 0.031 | 0.068 | 0.043 | −0.035 | 0.016 | −0.051 | −0.005 | 0.064 | −0.019 | 0.002 | 0.005 | 0.082 * | 0.041 |
| | 33 | Thunder | 0.016 | 0.090 * | 0.078 * | 0.048 | 0.067 | −0.026 | −0.012 | −0.038 | −0.008 | 0.035 | −0.032 | 0.019 | 0.030 | 0.017 | 0.006 |
| | 34 | Thunderstorm | 0.029 | 0.027 | 0.040 | 0.086 * | 0.083 * | −0.016 | −0.036 | −0.024 | −0.008 | 0.060 | −0.003 | −0.019 | 0.026 | 0.080 * | 0.026 |
| Musical Sound | 35 | Instrumental | −0.086 * | −0.112 ** | −0.093 * | −0.135 ** | −0.145 ** | 0.120 ** | 0.029 | 0.041 | −0.014 | −0.058 | 0.075 * | 0.032 | −0.064 | −0.046 | −0.106 ** |
| | 36 | Vocal | −0.031 | −0.085 * | −0.109 ** | −0.102 ** | −0.116 ** | 0.094 * | 0.025 | 0.039 | −0.010 | 0.005 | 0.052 | −0.006 | −0.056 | −0.069 | −0.037 |
| | 37 | Electronic | −0.038 | −0.112 ** | −0.110 ** | −0.109 ** | −0.088 * | 0.081 * | 0.062 | 0.044 | 0.019 | −0.054 | 0.092 * | 0.051 | −0.053 | −0.049 | −0.081 * |
| Natural Sound | 38 | Leaves | −0.094 * | −0.111 ** | −0.201 ** | −0.244 ** | −0.158 ** | 0.047 | 0.046 | 0.054 | −0.035 | −0.009 | 0.095 * | 0.047 | −0.093 * | −0.161 ** | −0.102 ** |
| | 39 | Falling stone | −0.054 | −0.120 ** | −0.137 ** | −0.191 ** | −0.120 ** | 0.062 | −0.005 | 0.032 | −0.006 | −0.043 | 0.071 | 0.006 | −0.036 | −0.126 ** | −0.072 |
| | 40 | Flying dust | −0.082 * | −0.133 ** | −0.156 ** | −0.186 ** | −0.146 ** | 0.074 * | 0.051 | 0.056 | −0.007 | −0.079 * | 0.079 * | 0.062 | −0.049 | −0.111 ** | −0.101 ** |
| | 41 | Flowing water | −0.062 | −0.144 ** | −0.143 ** | −0.198 ** | −0.106 ** | 0.066 | 0.031 | 0.034 | −0.008 | −0.034 | 0.059 | 0.012 | −0.039 | −0.127 ** | −0.103 ** |

Table 6. *Cont.*

| Sound Category | Code | Sound Source | Q21: Which Environment Do You Prefer to Stay in the Forest Park? | | | | | | | | | | | Q14: How Do You Get to the Forest Park Now? | | | |
|---|---|---|---|---|---|---|---|---|---|---|---|---|---|---|---|---|---|
| | | | In Sunlit Areas | In the Shade of Trees | Near a Snack Stall | In Secluded Areas | On the Water-side | In Lounge Areas | In Open Pave-ments | On Roadside Seats | In Higher Places | In Park Build-ings | On the Open Lawn | On Foot | By Bus | By Car | By Bike |
| | 42 | Mechanical | 0.086 * | 0.113 ** | 0.130 ** | 0.019 | −0.022 | −0.027 | −0.004 | −0.044 | 0.054 | −0.026 | −0.051 | −0.023 | 0.046 | 0.043 | 0.024 |
| | 43 | Construction site | 0.095 * | 0.087 * | 0.102 ** | 0.045 | −0.012 | −0.015 | −0.014 | −0.049 | 0.044 | −0.010 | −0.024 | −0.024 | 0.009 | 0.055 | 0.003 |
| | 44 | Handwork | 0.095 * | 0.080 * | 0.094 * | 0.013 | 0.028 | −0.021 | −0.021 | −0.024 | 0.057 | −0.024 | −0.037 | −0.023 | 0.053 | 0.019 | 0.026 |
| | 45 | Human activities | 0.084 * | 0.077 * | 0.097 ** | 0.030 | 0.036 | −0.075 * | −0.005 | −0.057 | 0.017 | −0.023 | −0.075 * | −0.027 | 0.117 ** | 0.050 | 0.010 |
| Other Sound | 46 | Mobile ringtones | 0.122 ** | 0.055 | 0.133 ** | 0.040 | −0.005 | −0.028 | −0.006 | −0.042 | 0.046 | −0.031 | −0.036 | −0.030 | 0.051 | 0.027 | 0.030 |
| | 47 | Children playing | 0.076 * | 0.080 * | 0.105 ** | 0.066 | 0.025 | −0.067 | −0.019 | −0.053 | 0.013 | −0.027 | −0.041 | −0.051 | 0.064 | 0.056 | 0.053 |
| | 48 | Street performance | 0.109 ** | 0.088 * | 0.090 * | 0.047 | 0.043 | −0.045 | −0.002 | −0.029 | −0.001 | 0.046 | −0.047 | 0.014 | 0.010 | 0.042 | 0.005 |
| | 49 | Sneezing | 0.051 | 0.089 * | 0.089 * | 0.016 | −0.013 | −0.017 | −0.046 | −0.035 | 0.028 | 0.011 | −0.071 | −0.017 | 0.028 | 0.067 | 0.021 |
| | 50 | Nonlocal dialect | 0.100 ** | 0.110 ** | 0.072 | 0.134 ** | 0.064 | −0.061 | −0.003 | −0.073 | 0.071 | −0.004 | −0.008 | −0.061 | 0.080 * | 0.114 ** | 0.015 |

Note: *: $p \leq 0.05$, **: $p \leq 0.01$.

**Table 7.** Correlation between personal conditions and soundscape preference.

| Sound Category | Code | Sound Source | Q12 | Q13 | Question 15 | | | | |
|---|---|---|---|---|---|---|---|---|---|
| | | | | | Exercise | Dog walking | Playing Chess | Square Dancing | Socializing |
| Vehicle Sound | 1 | Car | 0.178 ** | 0.192 ** | −0.041 | 0.074 * | 0.092 * | 0.068 | −0.044 |
| | 2 | Bus | 0.154 ** | 0.158 ** | −0.037 | 0.047 | 0.064 | 0.052 | −0.013 |
| | 3 | Express train | 0.152 ** | 0.121 ** | 0.011 | 0.010 | 0.011 | 0.023 | −0.062 |
| | 4 | Aircraft | 0.132 ** | 0.153 ** | −0.048 | 0.025 | 0.057 | 0.043 | −0.037 |
| | 5 | Fighter | 0.144 ** | 0.150 ** | −0.009 | 0.001 | −0.013 | 0.050 | 0.016 |
| | 6 | Motorcycle | 0.028 | 0.102 ** | −0.019 | 0.006 | 0.013 | 0.016 | −0.036 |
| | 7 | Tractor | 0.173 ** | 0.147 ** | −0.057 | 0.075 * | 0.110 ** | 0.075 * | −0.053 |
| | 8 | Bicycle | 0.105 ** | 0.144 ** | −0.007 | 0.007 | 0.022 | 0.072 | −0.072 |
| | 9 | Truck | 0.091 * | 0.070 | −0.067 | 0.014 | 0.028 | 0.092 * | −0.050 |
| | 10 | Police siren | 0.122 ** | 0.183 ** | −0.029 | 0.038 | 0.062 | 0.030 | −0.032 |
| | 11 | Ambulance siren | 0.144 ** | 0.180 ** | −0.028 | 0.134 ** | 0.099 ** | 0.022 | −0.057 |
| | 12 | Fire engine siren | 0.117 ** | 0.125 ** | −0.008 | 0.058 | 0.099 ** | 0.023 | −0.048 |
| Bird Song | 13 | Pigeon | −0.211 ** | −0.303 ** | 0.049 | −0.155 ** | −0.182 ** | −0.102 ** | 0.035 |
| | 14 | Wild goose | −0.183 ** | −0.233 ** | 0.038 | −0.139 ** | −0.185 ** | −0.073 * | 0.015 |
| | 15 | Swallow | −0.101 ** | −0.255 ** | 0.009 | −0.159 ** | −0.157 ** | −0.054 | 0.042 |
| | 16 | Eagle | −0.162 ** | −0.235 ** | −0.011 | −0.123 ** | −0.148 ** | −0.060 | 0.013 |
| | 17 | Hawk | −0.170 ** | −0.224 ** | 0.024 | −0.175 ** | −0.198 ** | −0.034 | −0.015 |
| | 18 | Swan | −0.155 ** | −0.262 ** | 0.028 | −0.100 ** | −0.164 ** | −0.024 | 0.049 |
| | 19 | Egret | −0.183 ** | −0.251 ** | 0.012 | −0.103 ** | −0.148 ** | −0.043 | −0.004 |
| | 20 | Sparrow | −0.147 ** | −0.189 ** | 0.041 | −0.125 ** | −0.161 ** | −0.064 | 0.019 |
| Livestock Sound | 21 | Cattle | −0.254 ** | −0.219 ** | 0.111 ** | −0.224 ** | −0.127 ** | −0.056 | 0.028 |
| | 22 | Horse | −0.202 ** | −0.199 ** | 0.047 | −0.141 ** | −0.076 * | −0.047 | 0.031 |
| | 23 | Sheep/Goat | −0.202 ** | −0.199 ** | 0.087 * | −0.164 ** | −0.080 * | −0.073 * | 0.032 |
| | 24 | Chicken | −0.222 ** | −0.170 ** | 0.073 * | −0.143 ** | −0.088 * | −0.063 | 0.025 |
| | 25 | Dog | −0.228 ** | −0.151 ** | 0.110 ** | −0.185 ** | −0.100 ** | −0.068 | 0.014 |
| | 26 | Pig | −0.216 ** | −0.184 ** | 0.151 ** | −0.144 ** | −0.106 ** | −0.086 * | 0.001 |
| | 27 | Duck | −0.213 ** | −0.127 ** | 0.070 | −0.184 ** | −0.116 ** | −0.035 | 0.034 |
| | 28 | Cat | −0.251 ** | −0.189 ** | 0.065 | −0.162 ** | −0.084 * | −0.038 | −0.042 |
| | 29 | Goose | −0.202 ** | −0.136 ** | 0.071 | −0.156 ** | −0.094 * | −0.043 | 0.023 |
| Atmospheric Sound | 30 | Rain | 0.182 ** | 0.209 ** | −0.100 ** | 0.220 ** | 0.126 ** | 0.032 | 0.043 |
| | 31 | Wind | 0.110 ** | 0.133 ** | −0.126 ** | 0.172 ** | 0.095 * | 0.019 | 0.046 |
| | 32 | Snow | 0.151 ** | 0.104 ** | −0.081 * | 0.074 * | 0.052 | 0.076 * | 0.055 |
| | 33 | Thunder | 0.070 | 0.111 ** | −0.059 | 0.109 ** | 0.053 | 0.038 | 0.056 |
| | 34 | Thunderstorm | 0.131 ** | 0.195 ** | −0.049 | 0.109 ** | 0.066 | 0.015 | 0.020 |
| Musical Sound | 35 | Instrumental | −0.195 ** | −0.248 ** | 0.066 | −0.142 ** | −0.116 ** | −0.076 * | −0.022 |
| | 36 | Vocal | −0.133 ** | −0.219 ** | 0.073 * | −0.121 ** | −0.095 * | −0.046 | −0.026 |
| | 37 | Electronic | −0.124 ** | −0.193 ** | 0.085 * | −0.116 ** | −0.093 * | −0.057 | −0.041 |
| Natural Sound | 38 | Leaves | −0.287 ** | −0.306 ** | 0.083 * | −0.150 ** | −0.184 ** | −0.076 * | −0.039 |
| | 39 | Falling stone | −0.254 ** | −0.290 ** | 0.093 * | −0.141 ** | −0.179 ** | −0.110 ** | −0.009 |
| | 40 | Flying dust | −0.241 ** | −0.248 ** | 0.060 | −0.092 * | −0.158 ** | −0.047 | −0.051 |
| | 41 | Flowing water | −0.245 ** | −0.248 ** | 0.085 * | −0.162 ** | −0.198 ** | −0.070 | −0.016 |
| Other Sound | 42 | Mechanical | 0.139 ** | 0.173 ** | −0.034 | 0.077 * | 0.098 ** | 0.003 | 0.030 |
| | 43 | Machine noise | 0.127 ** | 0.167 ** | 0.003 | 0.099 ** | 0.076 * | 0.010 | 0.014 |
| | 44 | Construction noise | 0.051 | 0.137 ** | −0.008 | 0.123 ** | 0.118 ** | −0.044 | −0.019 |
| | 45 | Exercise sound | 0.077 * | 0.156 ** | 0.000 | 0.070 | 0.079 * | 0.014 | 0.021 |
| | 46 | Mobile ringtones | 0.105 ** | 0.123 ** | −0.075 * | 0.069 | 0.084 * | −0.007 | 0.014 |
| | 47 | Children playing | 0.144 ** | 0.148 ** | −0.075 * | 0.079 * | 0.115 ** | 0.024 | 0.049 |
| | 48 | Footstep | 0.105 ** | 0.192 ** | −0.063 | 0.115 ** | 0.119 ** | 0.028 | 0.019 |
| | 49 | Vehicle noise | 0.143 ** | 0.117 ** | −0.048 | 0.035 | 0.077 * | 0.047 | 0.021 |
| | 50 | Bus noise | 0.178 ** | 0.192 ** | 0.001 | −0.246 ** | −0.260 ** | −0.303 ** | −0.253 ** |

Note: *: $p \leq 0.05$, **: $p \leq 0.01$. Q12: How long does it take you to get to the Forest Park from where you live? Q13: How often do you go to the Forest Park? Q15: What's your purpose of going to the Forest Park?

### 3.4. Participants

### 3.4.1. Gender

The results (as shown in Figure 5) showed that male and female older adults had the same average soundscape preferences. Among the four sound categories of vehicle sound, bird song, atmospheric sound, and other sound, men had a higher average soundscape preference than women.

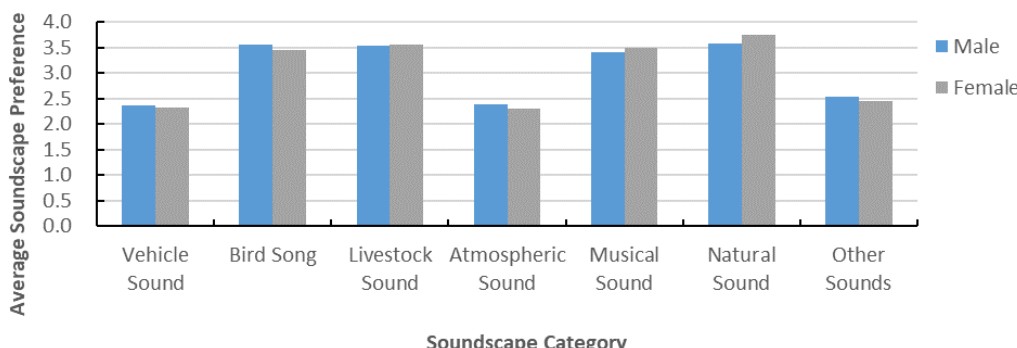

**Figure 5.** Average soundscape preferences of different genders.

### 3.4.2. Age

The results (as shown in Figure 6) showed that older adults of different ages had different preferences for different sound categories. Overall, the mean values of soundscape preference in all seven sound categories were higher for the individuals aged 80–84 years than for the other age groups. In contrast, the mean values of soundscape preference in the above seven sound categories were smaller for individuals aged 60–64 years than for other age groups.

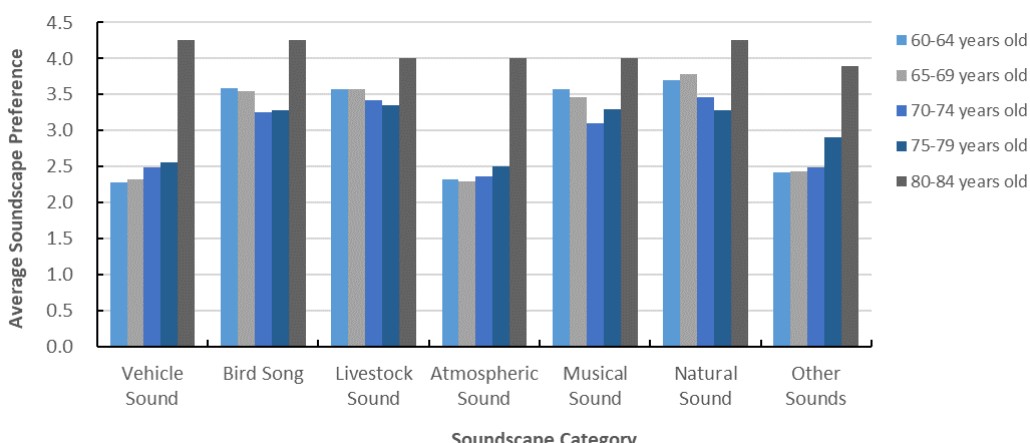

**Figure 6.** Average soundscape preference with different age groups.

### 3.4.3. Occupation

The results (as shown in Figure 7) show that older adults from different occupations had different preferences for different sound categories.

In the four sound categories of bird sound, livestock sound, musical sound, and natural sound, older adults from service worker occupations had higher soundscape preferences. In the three sound categories of vehicle sound, atmospheric sound, and other sound, older adults with the occupation of government officials and business managers had higher mean values of soundscape preference.

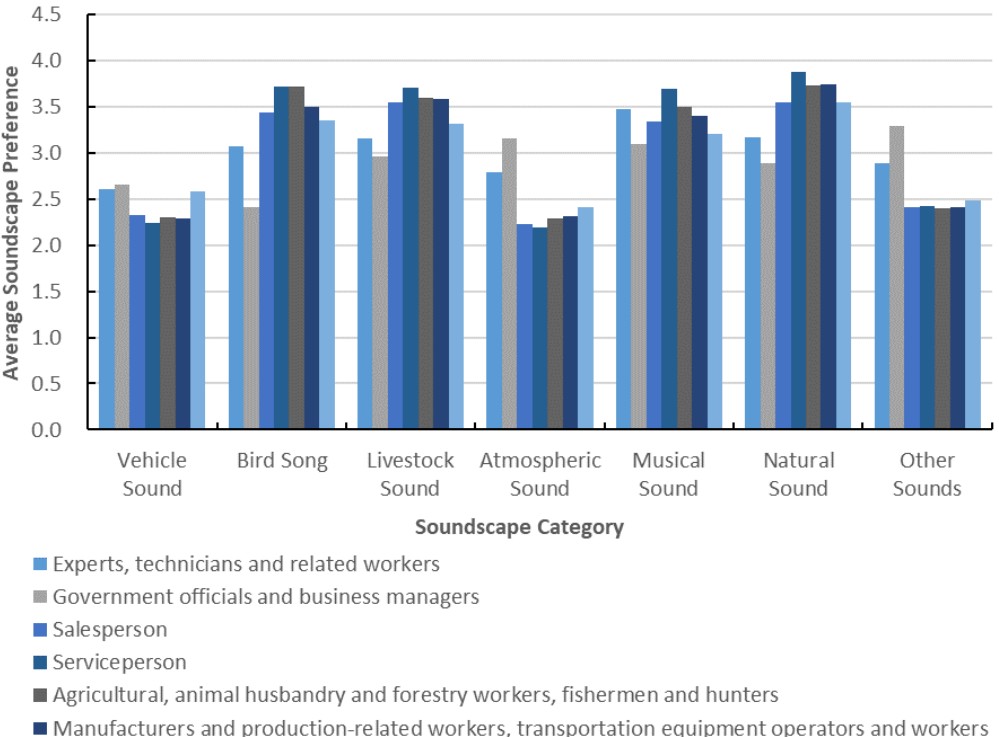

**Figure 7.** Average soundscape preferences with varying occupations.

### 3.4.4. Education Background

The results (as shown in Figure 8) showed that older adults with different educational backgrounds had different preferences for different sound categories. Figure 8 showed that across the seven sound categories, older adults with educations of a master's degree or higher had higher mean values of soundscape preference.

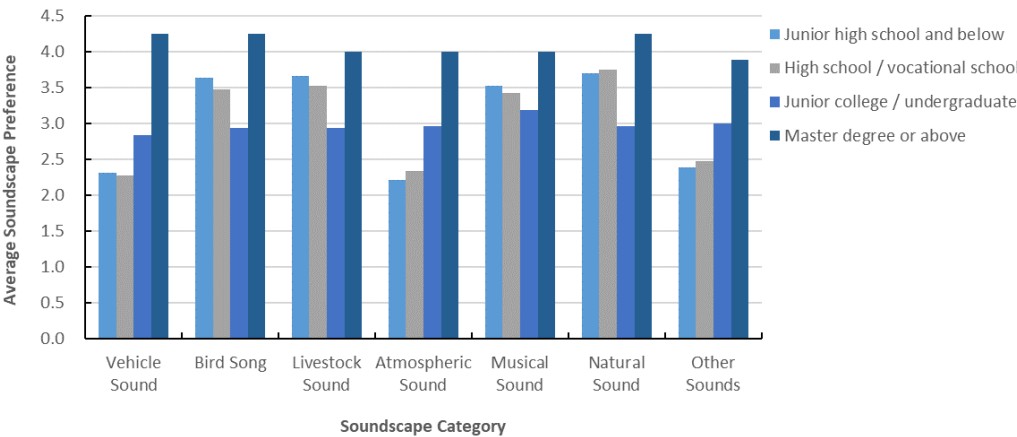

**Figure 8.** Average soundscape preferences with varying education backgrounds.

### 3.4.5. Living Conditions

The results (Figure 9) showed that older adults with different living conditions had different preferences for different sound categories.

Figure 9 also showed that in the four sound categories of bird song, livestock sound, musical sound, and natural sound, older adults living with a partner had higher mean values of soundscape preference. In the three sound categories of vehicle sound, atmospheric sound and other sounds, elderly people living in homes had higher mean values of soundscape preference.

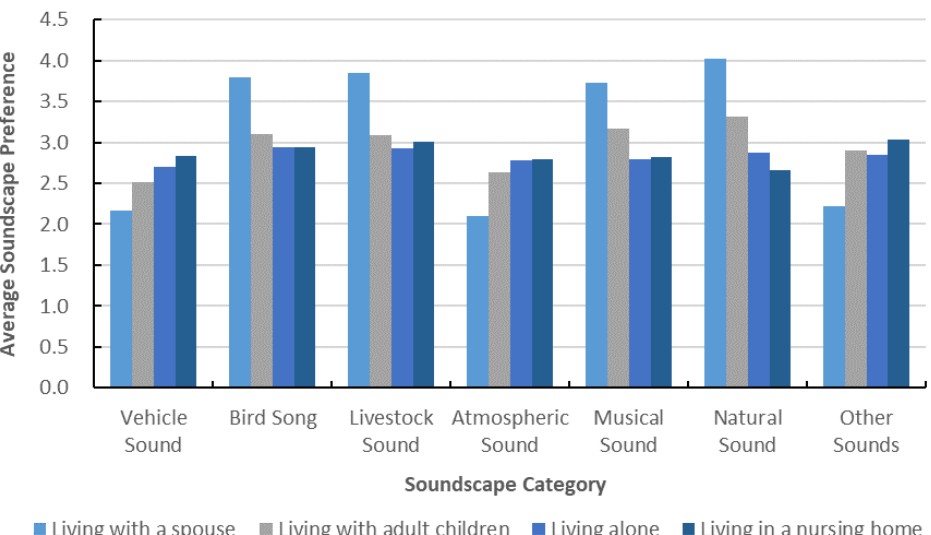

**Figure 9.** Average soundscape preferences with varying living conditions.

## 4. Regression Coefficient Model of Soundscape Preference

To further analyze the soundscape preferences, a regression coefficient model was applied with SPSS23.0, and the soundscape preference evaluation was divided into the target variables in Figure 10.

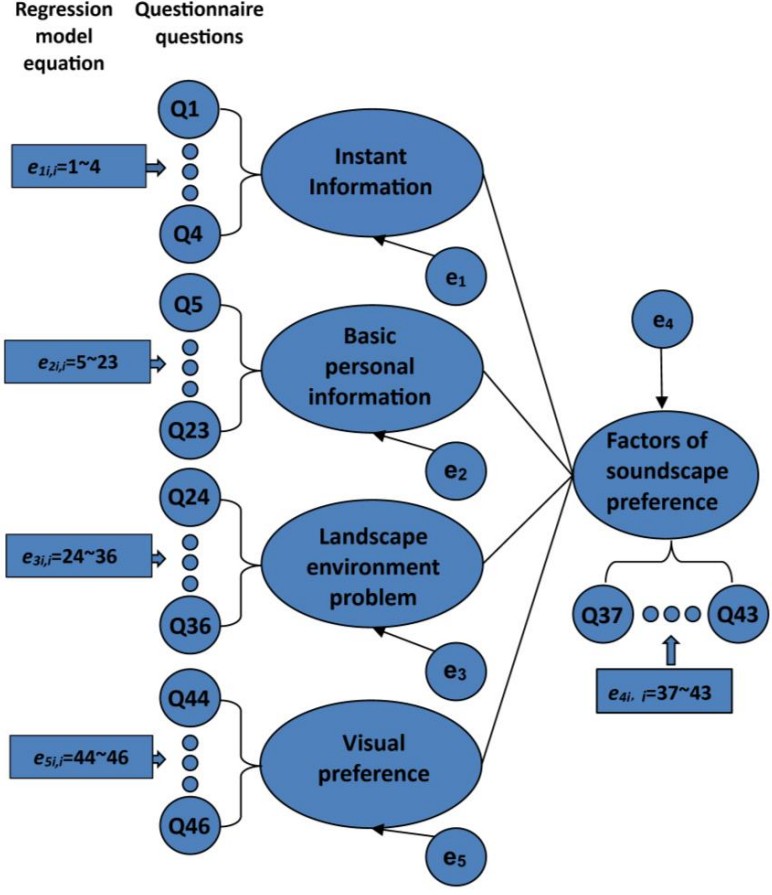

**Figure 10.** Structural equation model regression coefficients.

In this paper, the instant information, such as the weather condition, mood (Q1–Q4), basic personal information (Q5–Q23), landscape problem (Q24–Q36), and visual preference (Q44–Q46) on the day of visiting the forest park were used as independent variables, and

soundscape preference was used as the dependent variable to construct a regression model. Considering the possible problem of covariance among the explanatory variables, the optimal model was constructed via stepwise regression, and the results of the stepwise regression are shown in Table 8.

**Table 8.** Results of the stepwise regression.

| Model | R | R Square | Adjusted R Square | Std. Error of the Estimate | Durbin–Watson |
|---|---|---|---|---|---|
| 1 | 0.272 a | 0.074 | 0.073 | 0.373 | |
| 2 | 0.322 b | 0.104 | 0.101 | 0.367 | |
| 3 | 0.345 c | 0.119 | 0.115 | 0.365 | |
| 4 | 0.363 d | 0.132 | 0.127 | 0.362 | |
| 5 | 0.377 e | 0.142 | 0.136 | 0.360 | |
| 6 | 0.389 f | 0.152 | 0.145 | 0.358 | |
| 7 | 0.399 g | 0.160 | 0.151 | 0.357 | |
| 8 | 0.406 h | 0.165 | 0.155 | 0.356 | |
| 9 | 0.412 i | 0.170 | 0.159 | 0.356 | 1.865 |

(a) Predictors: (constant); (b) predictors: (constant), Q28; (c) predictors: (constant), Q28, and Q16C; (d) predictors: (constant), Q28, Q16C, and Q13; (e) predictors: (constant), Q28, Q16C, Q13, and Q461; (f) predictors: (constant), Q28, Q16C, Q13, Q461, and Q21E; (g) predictors: (constant), Q28, Q16C, Q13, Q461, Q21E, and Q18A; (h) predictors: (constant), Q28, Q16C, Q13, Q461, Q21E, Q18A, and Q29; (i) predictors: (constant), Q28, Q16C, Q13, Q461, Q21E, Q18A, Q29, and Q20C.

Subsequently, an analysis of variance (ANOVA) was performed on the data, and the results obtained are shown in Table 9 below.

**Table 9.** Results of ANOVA.

| | Model | Sum of Squares | df | Mean Square | F | Sig. |
|---|---|---|---|---|---|---|
| 1 | Regression | 8.039 | 1 | 8.039 | 57.701 | 0.000 a |
| | Residual | 100.723 | 723 | 0.139 | | |
| | Total | 108.761 | 724 | | | |
| 2 | Regression | 11.257 | 2 | 5.628 | 41.678 | 0.000 b |
| | Residual | 97.504 | 722 | 0.135 | | |
| | Total | 108.761 | 724 | | | |
| 3 | Regression | 12.942 | 3 | 4.314 | 32.460 | 0.000 c |
| | Residual | 95.820 | 721 | 0.133 | | |
| | Total | 108.761 | 724 | | | |
| 4 | Regression | 14.314 | 4 | 3.579 | 27.280 | 0.000 d |
| | Residual | 94.447 | 720 | 0.131 | | |
| | Total | 108.761 | 724 | | | |
| 5 | Regression | 15.434 | 5 | 3.087 | 23.782 | 0.000 e |
| | Residual | 93.327 | 719 | 0.130 | | |
| | Total | 108.761 | 724 | | | |
| 6 | Regression | 16.499 | 6 | 2.750 | 21.400 | 0.000 f |
| | Residual | 92.262 | 718 | 0.128 | | |
| | Total | 108.761 | 724 | | | |
| 7 | Regression | 17.355 | 7 | 2.479 | 19.447 | 0.000 g |
| | Residual | 91.407 | 717 | 0.127 | | |
| | Total | 108.761 | 724 | | | |
| 8 | Regression | 17.914 | 8 | 2.239 | 17.649 | 0.000 h |
| | Residual | 90.847 | 716 | 0.127 | | |
| | Total | 108.761 | 724 | | | |
| 9 | Regression | 18.459 | 9 | 2.051 | 16.240 | 0.000 i |
| | Residual | 90.302 | 715 | 0.126 | | |
| | Total | 108.761 | 724 | | | |

(a) Predictors: (constant); (b) predictors: (constant), Q28; (c) predictors: (constant), Q28, and Q16C; (d) predictors: (constant), Q28, Q16C, and Q13; (e) predictors: (constant), Q28, Q16C, Q13, and Q461; (f) predictors: (constant), Q28, Q16C, Q13, Q461, and Q21E; (g) predictors: (constant), Q28, Q16C, Q13, Q461, Q21E, and Q18A; (h) predictors: (constant), Q28, Q16C, Q13, Q461, Q21E, Q18A, and Q29; (i) predictors: (constant), Q28, Q16C, Q13, Q461, Q21E, Q18A, Q29, and Q20C.

As it can be seen from Table 10, the stepwise regression showed that there were nine regression models to be constructed, among which the R-squared (0.170) and adjusted R-squared (0.159) values of model 9 had the largest values among the nine models and

the standard error of estimation (0.35538) had the smallest value; therefore, model 9 was selected as the optimal regression model in this paper, and a subsequent analysis was conducted.

**Table 10.** Results of regression coefficient model.

| Model 9 | B | Std. Error | Beta | t | Sig. | Tolerance | VIF |
|---|---|---|---|---|---|---|---|
| (Constant) | 3.203 | 0.072 | | 44.752 | 0.000 | | |
| Q11 | −0.045 | 0.017 | −0.109 | −2.676 | 0.008 | 0.700 | 1.429 |
| Q28 | −0.039 | 0.019 | −0.083 | −2.051 | 0.041 | 0.709 | 1.410 |
| Q16C | −0.090 | 0.039 | −0.087 | −2.303 | 0.022 | 0.809 | 1.237 |
| Q13 | −0.044 | 0.014 | −0.117 | −3.082 | 0.002 | 0.810 | 1.235 |
| Q46_1 | 0.039 | 0.012 | 0.118 | 3.237 | 0.001 | 0.880 | 1.136 |
| Q21E | −0.174 | 0.068 | −0.095 | −2.578 | 0.010 | 0.851 | 1.174 |
| Q18A | 0.071 | 0.027 | 0.091 | 2.654 | 0.008 | 0.993 | 1.007 |
| Q29 | −0.050 | 0.024 | −0.074 | −2.116 | 0.035 | 0.948 | 1.055 |
| Q20C | −0.069 | 0.033 | −0.075 | −2.077 | 0.038 | 0.900 | 1.112 |

Dependent variable: soundscape preference.

Table 10 showed that the F-test statistic for model 9 was 16.240, with a significance value, *p*, of less than 0.001, indicating that there was a significant linear relationship between the independent variables.

As can be seen from Table 9, Model 9 contained a total of nine independent variables, namely Q11 (residence status), Q28 (what kind of sky you prefer to see), Q16C (coming to the forest park at noon), Q13 (frequency of visiting the forest park), Q46_1 (the forest street interface gives you a familiar feeling), Q21E (preferring to stay at the waterfront environment in the forest park), Q18A (placing more importance on road signs in forest parks), Q29 (hearing), and Q20C (placing more importance on the science education exhibition function of buildings in forest parks). The non-standardized regression coefficients of these nine independent variables were −0.045, −0.039, −0.090, −0.044, 0.039, −0.174, 0.071, −0.050, and −0.069, respectively. The regression coefficients of these nine independent variables were significant at the 5% level of significance, indicating a significant effect on soundscape preference for all of them. Among them, two independent variables, Q46_1 (forest street interface gives you a familiar feeling) and Q18A (more emphasis on road signs in forest parks), had positive effects on soundscape preference, while the rest of the factors had negative effects. Table 9 also shows that the tolerance values for each independent variable in model 9 were greater than 0.10 and the VIFs were less than 10, indicating that there was no multiple covariance problem.

The final regression model is summarized as:

$$y = 3.203 - 0.045*Q11 - 0.039*Q28 - 0.09*Q16C - 0.044*Q13 + 0.039*Q461 - 0.174*Q21E + 0.071*Q18A - 0.050*Q29 - 0.069*Q20C.$$

## 5. Discussion

In this study, an online questionnaire survey was conducted using a subjective evaluation method in the vicinity of Maoershan National Forest Park in Yanji, China. The data were analyzed and organized to understand the preferences of elderly people for forest soundscapes and the association between the soundscape preferences of elderly people and landscape characteristics. The investigation of soundscapes provides us with a new perspective for the development of urban forest parks [54]. The main findings are summarized as follows.

For the elderly, the preference for various sound sources in descending order is natural sound, animal sound, bird song, musical sound, vehicle sound, and atmospheric sound. In other words, compared with other sound sources, natural sound have an important influence on the elderly. Similarly, according to previous research, people prefer natural sound and most sounds associated with human activities [55]. However, they tend to

dislike mechanical sound. As for the soundscape of forest parks, the main preferences of the elderly are the sound of leaves, the sound of falling stones, the sound of dust rising, the sound of tinkling brooks and the sound of birds.

In view of the preferences of the elderly for natural sound, the government can incorporate natural sound elements such as bird songs, wind sound, and water flowing sound into urban forest parks. Additionally, diverse soundscapes can be provided, such as music performance areas or musical fountains, to cater to the different preferences of the elderly. This can be achieved through well-planned and designed vegetation, water features, and landscapes in the park, creating a harmonious and pleasant acoustic environment.

Regarding the influences of various respondent characteristics on soundscape preference, the difference in gender was not statistically significant, and the average soundscape preferences of elderly men and elderly women are very similar; elderly people of different ages have different preferences for different sound categories; elderly people of different occupations have different preferences for different sound categories, and the those occupied as government officials and business managers have different preferences for different sound categories. The mean value of soundscape preference was higher for the elderly individuals with employment as government officials and business managers; the mean value of soundscape preference was higher for the elderly individuals with different educational backgrounds, and the mean value of soundscape preference was higher for the elderly individuals with master's degrees or above; the mean value of soundscape preference was higher for the elderly individuals with different living conditions, and the mean value of soundscape preference was higher for the elderly individuals living in homes.

Taking into consideration the preferences and needs of the various elderly individuals for soundscapes, policymakers can introduce interactive landscape elements in urban forest parks. This may include interactive musical installations, sound sculptures, or participatory music activities that have therapeutic qualities. These interactive landscapes can provide a sense of engagement and enjoyment for the elderly, enhancing their interaction with the sound environment and creating a positive acoustic environment and community atmosphere for them.

The regression equation model established in this study revealed that among the various factors influencing the soundscape preferences of the elderly, the top five most influential independent variables were whether they liked to stay in the waterfront environment in the forest park, whether they came to the forest park at noon, whether they valued the road signs in the forest park, whether they valued the science education exhibition function of the buildings in the forest park, and the auditory situation. In light of the elderly people's preferences for each sound source, more water-related natural environments such as artificial rivers should be built in the park.

## 6. Conclusions, Reflection, Limitations and Future Work

Although previous research has attempted to explore the relationship between the sensory perceptions and behavioral experiences of people in urban parks, few studies have approached the issue from the perspective of older adults, especially those in underdeveloped areas. In this study, we used a subjective evaluation method, namely, a questionnaire survey on Maoershan National Forest Park, to explore the relationships between older adults' preferences for urban forest park soundscapes in underdeveloped cities and made recommendations for urban forest park design in accordance with the findings of the survey and relevant theoretical foundations.

### 6.1. Conclusions

6.1.1. Landscape Design Recommendations to Enhance the Soundscape Experience

(1)    Overall Landscape Design

It was found that the subjective evaluation of a soundscape is closely related to landscape design. The results of the questionnaire showed that the elderly had higher

preferences for the natural environment and the sounds of birds and animals. Older people of different genders, educational backgrounds, and physical qualities had different preferences for different soundscape elements. It is suggested that to cater to the preferences of each group, the integration of the soundscape should be fully considered in the overall landscape design to create a good layout. A harmonious environment can be created through the architectural design of the landscape structure, water features, plant design, etc., and a good audiovisual environment can be created by integrating soundscapes and different landscapes.

(2)    Green Environment Design

It was found that the degree of park greenery and the purpose and frequency of public green space use influenced the soundscape preferences of older adults. For example, as shown in Table 5, the mean values of the older adults' preferences for soundscapes were higher in environments with shade trees and open lawns. Previous studies have also found that the increased exposure of older adults to greenery can reduce mental stress and thus influence soundscape evaluations. Green landscaping on sidewalks and trails can produce sounds such as wind blowing in the leaves, thus stimulating resonance with natural sound. However, it is not suitable to set green landscapes outside of sports and leisure facilities, such as promenades, sports grounds, and benches, which mainly emphasize the soundscape of human activities.

Meanwhile, reasonable planning of plant shapes and colors, plant effects, plant distribution, plant types and terrace design can be used to divide areas for the elderly. In addition, different plant zones can be set up for elderly people with different plant preferences and physical health conditions, and water features should also be added near the plant zones to play with aesthetics and adjust the microclimate of the area.

### 6.1.2. Soundscape Design for the Elderly

It can be inferred from the questionnaire results that the elderly, as a group with a more complex situation, also have greater differences in their physical condition, and the elderly individuals with different physical qualities have different hobbies and different patterns of participating in urban forest parks. In view of this observation, the following landscape suggestions are proposed.

It was found that the elderly individuals in better physical condition preferred to visit places with mountains, rivers and forests, and these individuals had higher preferences for complex landscapes. On the contrary, for the elderly individuals who were in poor physical condition or had physical disabilities, the results of the questionnaire showed that they did not have a great preference for exercising in urban forest parks and even had a lower preference for some landscapes with high activity requirements. However, they still enjoyed hearing crowd activities, especially the rhythm and melody of square dancing. This finding implies that overly complex landscapes become a burden for such elderly people. Therefore, under the premise of protecting the safety of the elderly and controlling cost, designers should design landscapes that the elderly like. For example, the complex landscape can be set far away from the intersection, and an area near the entrance and exit can be set up to ensure the sound reception of such elderly people as much as possible by equipping speakers with appropriate volume to ensure that it is not noisy. For people with hearing impairments, visual cues such as text, lights, and guardrails will ensure their safety and improve their viewing experience as much as possible. Forest parks designed and built with such considerations will meet their soundscape preferences and entice them to spend time the space, which will be beneficial to their physical and mental health and can also satisfy the mobility requirements of the elderly individuals who are more physically active.

To ensure consistently favorable sound environments in urban forest parks, the government should undertake regular maintenance and management work. This includes monitoring and controlling sources of noise pollution, maintaining park facilities and sound equipment in good condition, and promptly addressing any issues that may affect the soundscape environment.

By implementing these policy recommendations, urban forest parks can provide an ideal soundscape environment that meets the preferences of the elderly for natural sounds and creates positive sound experiences and interactive opportunities for them. This will surely contribute to enhancing their quality of life and promoting their physical and mental well-being.

### 6.2. Reflection

This study adopted a subjective evaluation method to investigate the soundscape preferences of the older residents of underdeveloped cities in China for urban forest parks. Though the method proved reliable and valid, the data were only statistically analyzed from a correlation perspective, and other possible factors, such as social environment, epidemic context, level of urban management, and level of urban infrastructure, were not taken into account. It is suggested that such factors be incorporated into future research so as to provide a profound theoretical basis for the construction of soundscapes in urban forest park environments.

Overall, this study was successful in identifying the influence of different factors on the soundscape preferences of older adults in urban forest parks. These findings have rich implications for park designers and managers in developing relevant design and management strategies to enhance the evaluation and perception of elderly people.

### 6.3. Limitations

This study focused on social hotspots and analyzed the soundscape preferences of the elderly in urban forest parks in underdeveloped cities. The limitations and advantages of the research are presented and implications for future research as follows. Although studies on soundscape preference typically focus on people's perceptions and evaluations of sounds in natural environments rather than solely quantifying noise, soundscape preference research aims to understand people's preferences and evaluations of different sound environments as well as the impact of these sounds on their emotions, cognition, and behavior. This study dealt with the soundscape of a forest park, with a focus on elderly people's perceptions of natural sound such as bird songs, wind rustling, and water flow. However, for a more comprehensive understanding of the soundscape, future research should consider specific measurements of noise in urban forest parks.

As noted previously, soundscape preference was indirectly affected by the epidemic; in fact, all aspects of thought and life have been affected by the epidemic. Thus, it should be noted that the conclusions of this study might have been skewed due to the indirect potential impact of the epidemic on soundscape preference.

### 6.4. Future Work

This study constructed a comprehensive and in-depth questionnaire to capture the influence of many aspects on older individuals' soundscape preferences and arrived at convincing conclusions, lending support to the findings of earlier investigations. But there are three works are expected to be refined in the future.

(1) To maximize the understanding of the influences on the soundscape preferences of older adults, the relevant literature was comprehensively and profoundly reviewed, which generated rich implications for the design and administration of the questionnaire.

(2) To determine the key factors that influence the soundscape preference of the elderly, a regression coefficient model and an automatic linear model were established which effectively guaranteed the accuracy of the data analysis and interpretation.

(3) To bring light to the construction and maintenance of forest parks aimed at improving the well-being of elderly people, well-grounded recommendations were provided for landscape designers on how to cater to the soundscape preferences of different elderly groups.

**Author Contributions:** Conceptualization, J.X., L.L. and Q.Z.; data curation, L.L.; formal analysis, Q.Z. and Y.M.; investigation, J.X. and L.L.; methodology, Q.Z. and L.L.; resources, J.X.; software, Y.M. and Y.P.; validation, Y.M. and Y.P.; writing—original draft, Q.Z. and L.L.; writing—review and editing, T.W. All authors have read and agreed to the published version of the manuscript.

**Funding:** This research was supported by China youth fund of national natural science foundation projects (52208057), the Guangdong Provincial Natural Science Foundation General Project (2023A1515011191), the Ministry of Education humanities social sciences research project (20YJC760101), the Zhejiang cultural relics protection science and technology project (2021017), the Zhejiang Provincial Philosophy and Social Sciences Planning Project Youth Fund (21NDQN213YB), and the Major humanities and social sciences research program of colleges and universities in Zhejiang (2021QN053).

**Data Availability Statement:** The data in this study are available from the authors upon request.

**Conflicts of Interest:** The authors declare no conflict of interest.

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
