# Peer review of "A Study on the Soundscape Preferences of the Elderly in the Urban Forest Parks of Underdeveloped Cities in China"

_forests, doi:10.3390/f14061266_

Round 1

Reviewer 1 Report

The authors dedicated their analysis to research on the soundscape preference of the elderly in urban forest parks in underdeveloped cities in China. Whereas the study is interesting, there is no explanation as to a study on the case of Maoershan Mountain National Forest Park in Yanji, Jilin; China was chosen. This choice should be part of the methodology section, with adequate analysis resolving particular choices. The reasons pointing out  - why - are presented, but the reader receives no further data concerning other possible choices leading to the final conclusion as to why this particular site is amongst other sites. The authors explained that the respondents were chosen at random + gave the number of responces, but there is no indication of the number of total users and, therefore no explanation as to the percentage of the total users which is presented for the survey. Further analysis is prepared. on a good level, but some additional explanation should be included - as stated above. The conclusion section should be placed at the end after the discussion section. Presently the discussion section does not present and links with other existing or foreseen research. The definition of "an urban forest" has not been included - I would advise the authors to search the contents of https://cities4forests.com/lg-urban-forests/what-is-an-urban-forest/ which might prove helpful. 

Moderate editing of English language required - I would advise the authors also  to slightly re-phrase the title, as the wording could be better.

Author Response

Thank you for your review, your comments are very important to us, and thank you for giving us the opportunity to submit changes. We have revised your comments as requested and have asked the professionals to revise them in English.

Reviewer 2 Report

The conclusions paragraph should be at the end of the paper.

The authors used a statistical approach, and as such can be influenced by how questions are asked.

Test results should be compared with those of other authors and any differences highlighted.

The sound levels (dBA, and frequency domain level, signal temporal history) in the park are not reported, so it is not possible to know the noise in the different areas, and possibly which areas are most popular.

A map is missing with indications of the routes followed and where people stop.

The number of people present in the park.

The approach with the tests is subjective, but should then be contextualized to the state of the park and how this is experienced.

It is not highlighted whether the people were trained for the tests.

It is not highlighted whether people have had experience with other parks.

It is not highlighted how to improve people's sound experience

Author Response

Thank you for your comments on our paper, your comments are very important to us, thank you for giving us the opportunity to submit revisions. We have revised the details one by one and submitted them in the attachment. Thank you again.

Reviewer 3 Report

The study examines the elderly population’s soundscape preference based on a survey. While the study sounds interesting, I have some comments to improve the quality as follows:

Abstract jumps to the methods so quickly. It needs some background information.

The study needs a robust introduction. It seems that it is a literature review in the current form.

The study should describe the elder population as it may differ for different countries.

Ln 122 and 130 seem irrelevant.

There are some typos/syntax/weird choice of words exists, particularly the last two paragraphs of the intro.

Ln 150 mathematical statistics?

Ln 153-156 seems irrelevant or should be placed somewhere earlier.

Figure 3 is very important; however, it does not read well. It needs to be recreated.

Table 4 should be condensed.

Is there any better way to create the figures (5-9) similar representation as they seem so different and make it difficult to follow?

Since the study did not include noise measurements, it can be mentioned as a limitation of the study in the related section.

The conclusion needs more policy and planning, even regulation aspects, rather than just summarizing the findings/results of the analyses.

What is the core message of the study? Most of the findings are already in the literature.

Mentioned in the comments. 

Author Response

Thank you for your heart for our papers, your comments are very important to us, thank you for giving us the opportunity to submit revisions. We have revised the details one by one and submitted them in the attachment. Thank you again.

Round 2

Reviewer 1 Report

The authors have responded to all of my queries. I believe the paper is presently ready to be published.

Reviewer 2 Report

accept

Reviewer 3 Report

Thank you for addressing my concerns.